# Psychometric validation of an Arabic version of the WHO-5 wellbeing index among Lebanese adolescents

Rita Doumit[1], Souheil Hallit[2,3,4], Maria-Jose Sanchez-Ruiz[5,6], Myriam El Khoury-Malhame 🆔[6]*

1 Alice Ramez School of Nursing, Lebanese American University, Beirut, Lebanon, 2 School of Medicine and Medical Sciences, Holy Spirit University of Kaslik, Jounieh, Lebanon, 3 Psychology Department, College of Humanities, Effat University, Jeddah, Saudi Arabia, 4 Applied Science Research Center, Applied Science Private University, Amman, Jordan, 5 Department of Personality Psychology, Assessment, and Psychological Treatment, School of Psychology, UNED, Madrid, Spain, 6 Department of Psychology and Education, Psychology Program, School of Arts and Sciences, Lebanese American University, Beirut, Lebanon

* myriam.malhame@lau.edu.lb

## Abstract

### Introduction

Wellbeing in adolescence is frequently associated with positive developmental outcomes. The WHO-5 is widely recognized for its brevity, clarity, robust reliability, and cultural validity. In this study, we aimed to assess the psychometric properties of an Arabic version of the WHO-5 scale among Lebanese adolescents.

### Methods

This cross-sectional study involved 700 Lebanese school students aged 14−19 years. Participants were assessed using WHO-5 wellbeing index (WHO-5) (wellbeing), PHQ-9 (depression), and GAD-7 (anxiety) Arabic questionnaires at two-time points, 3 months apart.

### Results

We found that the 5 items of the WHO-5 converged into a single factor. Internal consistency of scores was adequate in the total sample (ω = .83/ α = .83). The convergent validity for this model was satisfactory. We were able to show the invariance across gender at the configural, metric, and scalar levels, with males showing a higher level of wellbeing compared to females. The pre-posttest assessment for the WHO-5 scale was conducted on 358 participants; the intraclass correlation coefficient was adequate = 0.78 [95% CI .73; .82]. Our analyses also show that wellbeing was negatively correlated with depression (r = −.54; p < .001) and anxiety (r = −.52; p < .001).

**Data availability statement:** The authors do not have the right to share any data information since the data is owned by a third-party organization, the Lebanese American University. Data can be shared upon a reasonable request from Ms. Perla Zgheib (perla.z.zgheib@gmail.com), Lecturer, Department of Social and Education Sciences, School of Arts and Sciences, Lebanese American University, Beirut, Lebanon Other researchers can replicate the results we obtained by following the outlined methods described in the paper. Authors did not have special privileges beyond what was granted by the Lebanese American University. The data used can be accessed by other researchers after obtaining the approval of the Lebanese American University ethics committee.

**Funding:** This research was funded by Grand Challenges Canada – Global Mental Health – NIHS; R-GMH-POC-2107-47498. The funders had no role in study design, data collection and analysis, decision to publish, or preparation of the manuscript.

**Competing interests:** The authors have declared that no competing interests exist.

## Conclusion

The Arabic WHO-5 among Lebanese adolescents displayed highly satisfactory psychometric properties, which are evidence of its validity. It could be used to better track positive mental health in this vulnerable age group and could highlight efficiency in interventions aiming to promote wellbeing in adolescents. It could also potentially identify at risk individuals.

## Introduction

Wellbeing is a multifaceted construct mostly linked to positive mental health, emotional stability, and psychological wellness [1]. The World Health Organization emphasizes its definition as a "state of complete physical, mental and social wellbeing and not merely the absence of disease or infirmity" [2]. In the absence of unified definitions for wellbeing and with the lack of clear biomarkers, various scales have been validated to measure this construct. The most widely used versions include, yet are not restricted to, the World Health Organization Wellbeing Index (WHO-5) [3], General Wellbeing [4], the Wellbeing Numerical Rating Scales (WB-NRSs) [5] and Psychological Wellbeing Scale (PWB) measuring 6 aspects of wellbeing [6].

Other scales are known to assess conceptually similar constructs such as life satisfaction, positive affect, optimism, and happiness [7]. Among the validated wellbeing-related scales, the WHO-5 stands out for its brevity and ease of use, with only 5 items as opposed for instance to the PWB (42 items) and even its shorter version (18 items). It has strong psychometric properties, minimal overlap with scales measuring mental distress, and has also been translated into over 30 languages. Its adoption has been advocated by the WHO-led recommendations, with the aim of facilitating cross-cultural comparisons of the prevalence of mental wellness and distress [5]. Although this scale has been used in various nations and clinical settings, it has mostly been validated in high-income, western countries and only a few studies have targeted the Middle East [8] or particularly adolescents [9].

Adolescence is a critical developmental stage marked by blatant physical changes, driven by hormonal fluctuation, and often accompanied by emotional lability and interpersonal conflicts [10]. As young adolescents learn to navigate those changes, they become more susceptible to psychological distress and decreased wellbeing [11,12]. In this age group, there is a wealth of evidence linking wellbeing with various developmental benefits, including improved academic performance, reduced risks of mental health problems, stronger social ties, and increased resilience [13,14]. A few studies have investigated some of these benefits in Lebanese youth, especially those related to positive educational outcomes [15].

Measuring wellbeing using WHO-5 in the adolescent population have been conducted in Ghana [16], Europe [17], and ethnic minorities in Norway and Sweden [18]. In those single country studies, the WHO-5 wellbeing index has systematically shown a one factor structure similarly to the original version in adults with adequate convergent and divergent validity. Although a study involving 15 European countries had suggested

a better use of the WHO-4; a revisited version of the scale with adolescents [19] yet another very recent cross-national validation of the initial 5-items WHO-5 in adolescents conducted in 43 countries spanning altogether Europe, Central Asia and North America further documented the solid psychometric properties of this scale in more than 250,000 respondents aged 11–15 years [20]. Additionally, the scale has been validated in pediatric care and has shown good enough diagnostic accuracy as a screening tool for depression in adolescents [21]. However, despite its widespread use and promising robustness in the Global North mostly and in adult populations, there is a surprising gap in validation studies within the Arab region, which is known to host one of the most youthful age structures globally [14,22]. From a cultural perspective, adolescents' distress is particularly prominent in societies marked by collective traumas, often indicating an increased risk of psychiatric disorders as they transition into young adulthood [23]. Lebanon, as a small Middle Eastern country, bears the weight of accumulated national adversities stemming from decades of civil war, including severe socio-political unrest, the most devastating economic collapse of its modern history, and a cataclysmic explosion of the Beirut port. These challenges unfold against the backdrop of ongoing adjustments to the COVID-19 pandemic and the influx of nearly 1.5 million Syrian refugees [24]. Taken together with failing governmental institutions and a lack of mental health public policies, these crises have left the country with staggering psychological distress levels [25,26] and plummeting levels of wellbeing [27]. Recent studies particularly point to alarming levels of anxiety and depression in Lebanese adolescents reaching more than 60% for anxiety and 35% for depression after the Beirut blast [23], while others have focused on protective factors of wellbeing among Lebanese youth during the COVID-19 pandemic, such as trait Emotional Intelligence and meaning-centered coping [28].

In line with global concerns around adolescents' deteriorating mental health [29] and growing evidence of worsening their distress and isolation [28], determining levels of wellbeing in a vulnerable age group becomes of utmost importance, notably when taken against a backdrop of sociopolitical, healthcare, and financial instabilities. Given the dynamic nature of subjective wellbeing, it would allow the design of targeted interventions aimed at increasing its levels of wellbeing to bolster psychological distress. This study thus aims at validating a culturally-adapted Arabic version of the WHO-5 wellbeing index, a brief reliable instrument for wellbeing in adolescents. It contributes to assessing and subsequently promoting wellbeing in the future life trajectories of the Lebanese population and in the Arab-speaking context in general.

## Methods

### Procedure

The study was part of a larger project "Yes to Emotions in Youth (YEY): An Emotional Intelligence Training for Vulnerable Youth in Lebanon" funded by Grand Challenges Canada – Global Mental Health. It received administrative approval from the Lebanese Ministry of Education and Higher Education (MEHE) and was approved by the Institutional Review Board (LAU IRB; reference number: LAU.SAS.MJ1.28/Oct/2021), in accordance to the Declaration of Helsinki. The sample consisted of 700 high school students, aged between 14 and 19, who were recruited from four public schools in Lebanon, randomly assigned by MEHE, in September 1st 2022. Some participants were tested twice for the same scales, within 3 months, on the 15th of December 2022, to assess test-retest validity and reliability of the scales [30]. At retest, 358 participants opted to complete the survey and their responses were subsequently included. All participants and their parents signed the informed consent before enrolling in the study. Students within the appropriate age range who approved to enroll in the study were approached in class, in person, to fill the validated Arabic version (S1 Table) of the standardized scales for wellbeing (WHO-5), depression (PHQ-9), and anxiety (GAD-7), in that order. All scales are publicly available (S2 Table) and were presented to participants in the same order.

### Minimum sample size

We aimed to enroll a minimum of 100 adolescents following the recommendations of Mundfrom et al. of 3–20 times the number of the scale's variables [31] and those of Giner-Sorolla et al. [32].

## Questionnaires

**WHO-5 (World Health Organization Wellbeing Index; [33]).** It is a 5-item questionnaire measuring general emotional wellbeing in the past two weeks. Participants answer Likert-scale questions such as "I have felt cheerful and in good spirits" ranging from 0 "At no time" to 5 "All the time". Higher scores indicate higher states of wellbeing and are transformed to a 100-point basis. It has been shown to hold a reliability greater than ($\alpha = .8$) and has been validated in adult Arab-speaking population in six-countries [8]. The scale is openly available as per the link.

**GAD-7 (The Generalized Anxiety Disorder; [34]).** It is a self-reported questionnaire designed to assess anxiety during the previous 2 weeks. It is based on 7 items "e.g., Feeling nervous, anxious, or on edge", which are responded to on a 4-point Likert scale ranging from 0 "Not at all" to 3 "Nearly every day". Higher scores indicate higher severity of symptoms. It has satisfactory internal consistency ($\alpha$ from .79 –.91;) and has been validated among an Arabic Lebanese population [35]. The scale is openly available.

**PHQ-9 (Patient health questionnaire-9; [36]).** It is a 9-item questionnaire measuring the frequency of depressive symptoms over the past two weeks, with items such as having "I have little interest or pleasure in doing things" evaluated on a Likert-scale ranging from 0 "Not at all" to 3 "Nearly every day". Higher scores indicate higher severity of symptoms. It has strong reliability with ($\alpha = .82$), and was validated in the Lebanese population [35]. The scale is openly available.

## Analytic strategy

There were no missing responses in the dataset for the WHO-5 answers. However, some missing data was found in the other variables (age, gender, etc.) and were not replaced. We used data from the entire sample to perform a confirmatory factor analysis (CFA) using RStudio, "lavaan" and "SemTools" programs [37,38]. We used Weighted Least Squares with Mean and Variance (WLSMV) estimation method, which is more appropriate for ordinal data [39]. Parameter estimates were obtained using the maximum likelihood method. Multiple fit indices were calculated: root mean square error of approximation (RMSEA), standardized root mean square residual (SRMR), Tucker-Lewis Index (TLI), and Comparative Fit Index (CFI). Values ≤ .08 for RMSEA, ≤ .05 for SRMR, and ≥.90 for CFI and TLI indicate good fit of the model to the data [40]. The latent variable was scaled in the CFA model by fixing the loading of the first item to 1. Additionally, values of the average variance extracted (AVE) ≥.50 indicated evidence of convergent validity [41]. The assumption of multivariate normality was violated as indicated by Mardia's skewness = 139.83; $p < 0.001$ and Mardia's kurtosis = 3.98; $p < 0.001$, therefore parameter estimates were obtained using the robust maximum likelihood method. Despite non-normality, this estimator ensures a meaningful model fit evaluation by providing robust standard errors.

To examine gender and age invariance of WHO-5 scores, we conducted multi-group CFA [42] using the total sample. Age was dichotomized into middle adolescence (14−17 years) and late adolescence (18−19 years) [43]. Measurement invariance was assessed at the configural, metric, scalar and strict levels [44]. We accepted ΔCFI ≤ .010 and ΔRMSEA ≤ .015 or ΔSRMR ≤ .010 as evidence of invariance [45]. If measurement invariance was established, we planned on checking for a difference in WHO-5 total scores between males and females using the Mann-Whitney test.

Internal consistency was assessed using McDonald's ω and Cronbach's α, with values greater than.70 considered adequate [46,47]. Normality was not verified since the p value of the Shapiro-Wilk test turned out significant ($p < 0.001$). Spearman were calculated to examine the association between the WHO-5 scores and the other scales, with values of 0.2, 0.5 and 0.8 indicating weak, moderate and strong effect sizes respectively [48]. In order to assess the test-retest reliability of the WHO-5 wellbeing index, we calculated the intraclass correlation coefficient (ICC) using the reliability analysis option in SPSS v.27 to ensure the scale's stability over time. We used the two-way mixed-effect model ICC. The ICC values vary between 0 and 1, with values >0.75 considered excellent, good between 0.60 and 0.75, moderate between 0.40 and 0.60 and poor if < 0.40 [49].

 

# Results

## Participants

Seven hundred adolescents participated in this study. The mean and standard deviation of the scores were as follows: WHO-5 wellbeing index (pre: 41.75 ± 22.61, post: 39.68 ± 21.24), PHQ-9 depression (11.10 ± 6.76), and GAD-7 anxiety (9.62 ± 5.50). The description of the categorization of participants according to their PHQ-9 and GAD-7 scores is summarized in Table 1.

**Confirmatory factor analysis of the WHO-5 scale.** CFA indicated that fit of the one-factor model of the WHO-5 wellbeing index was excellent: Robust RMSEA = .044 (90% CI.028,.061), SRMR = .028, Robust CFI = .996, Robust TLI = .992. The standardized estimates of factor loadings were all adequate. Internal consistency of scores was adequate in the total sample ($\omega$ = .83/ $\alpha$ = .83). The convergent validity for this model was satisfactory, as AVE = .51. To evaluate local fit, we examined the modification indices between residuals; all values, including items 1–2 were below 10; therefore, we did not add a correlation between those two items.

**Gender invariance.** We were able to show the invariance across gender at the configural, metric, and scalar levels "S2 Table". Males showed a higher level of wellbeing compared to females (45.71 ± 23.84 vs 38.85 ± 21.08; $t$(671) = 3.96; $p$ < .001, Cohen's d = 0.307). No significant difference was found in terms of age categories (14–17 vs 18–19 years) (41.43 ± 22.50 vs 42.46 ± 23.80; $t$(652) = 0.39; $p$ = 0.699, Cohen's d = 0.045) (Table 2).

**Test-retest reliability: Intraclass correlation.** The pre-posttest for the WHO-5 scale was done on 358 participants. The intraclass correlation coefficient was excellent = 0.78 [95% CI .73; .82]. For the item-level test-retest reliability coefficients, the ICC values were as follows: item 1 (0.62 [95% CI .52; .69]), item 2 (0.61 [95% CI .51; .69]), item 3 (.65 [95% CI .57; .72]), item 4 (.63 [95% CI .54; .70]) and item 5 (.61 [95% CI .51; .69]). The correlation between the total scores pre and post was r = 0.64; p < 0.001 (moderate to strong effect).

**Concurrent validity.** Higher depression (rho = −.55; p < .001) and anxiety (rho = −.52; p < .001) were significantly associated with lower wellbeing.

**Discriminant validity.** Since the square root of the AVE (= 0.71) was higher than the correlations of the WHO-5 and the depression and anxiety ones, discriminant validity was verified.

**Table 1. Sociodemographic information and categorization of participants according to their depression and anxiety scores (N = 700).**

| Age (years) | 16.27 ± 1.11 |
|---|---|
| Gender | |
| Males | 295 (42.1%) |
| Females | 378 (54.0%) |
| Depression severity | |
| No depression (PHQ-9 scores 0–4) | 135 (19.3%) |
| Mild (PHQ-9 scores 5–9) | 193 (27.6%) |
| Moderate (PHQ-9 scores 10–14) | 146 (20.9%) |
| Moderately severe (PHQ-9 scores 15–19) | 130 (18.6%) |
| Severe (PHQ-9 scores 20–27) | 95 (13.6%) |
| Anxiety severity | |
| No anxiety (GAD-7 scores 0–4) | 147 (21.0%) |
| Mild (GAD-7 scores 5–9) | 226 (32.3%) |
| Moderate (GAD-7 scores 10–14) | 174 (24.9%) |
| Severe (GAD-7 scores ≥15) | 153 (21.9%) |

*Numbers might not round to the total N because of missing values.

**Table 2. Measurement invariance of the WHO-5 wellbeing in the total sample.**

| Model | CFI | RMSEA | SRMR | Model Comparison | ΔCFI | ΔRMSEA | ΔSRMR |
|---|---|---|---|---|---|---|---|
| Model 1: Gender invariance | | | | | | | |
| Configural | .970 | .102 | .027 | | | | |
| Metric | .973 | .083 | .029 | Configural vs metric | .003 | .019 | .002 |
| Scalar | .969 | .078 | .034 | Metric vs scalar | .004 | .005 | .005 |
| Strict | .971 | .067 | .034 | Scalar vs strict | .002 | .011 | <.001 |
| Model 2: Age invariance | | | | | | | |
| Configural | .981 | .083 | .023 | | | | |
| Metric | .983 | .067 | .026 | Configural vs metric | .002 | .016 | .003 |
| Scalar | .985 | .054 | .027 | Metric vs scalar | .002 | .013 | .001 |
| Strict | .989 | .042 | .027 | Scalar vs strict | .004 | .012 | <.001 |

CFI = Comparative fit index; RMSEA = root mean square error of approximation; SRMR = Standardized root mean square residual.

## Discussion

This study investigated the psychometric properties of the Arabic version of the WHO-5 wellbeing index in a group of Arabic-speaking Lebanese high-school students. Findings support a single-factor model, with adequate reliability (McDonald's $\omega$ = .83), as well as good convergent and concurrent validity. Measurement invariance across gender groups was also evidenced at the configural, metric, and scalar levels. Consequently, the WHO-5 could be exploited as a reliable and valid tool, time-effective and easy to use, with cultural sensitivity to evaluate Arab adolescents' overall wellbeing. Given that this scale has been mostly administered in adults in Western countries, investigating its validity among Lebanese adolescents further endorses its cross-cultural potential and its suitability for measurements of wellbeing in Arabic-speaking adolescents. This could also be particularly relevant for other countries sharing some of the overall settings [24], including vulnerable populations, low-income countries, and those plagued by atrocities of political unrest, ongoing wars, and/or severe monetary depreciation.

The WHO-5 scores in our sample of 681 school students were found to be lower compared to means documented in an international study among European adolescents with comparable age range and gender distribution [17]. Those averages were nonetheless comparable to those of young adults in Lebanon and in other low-middle-income Arab countries such as Morocco, Kuwait, and Jordan [8]. Taken together with values obtained on anxiety (GAD-7) and depression scales (PHQ-9), these results indicate elevated levels of mental distress among Lebanese-educated adolescents, with low levels of wellbeing, and heightened psychological distress in participants; reporting moderate to severe symptoms of anxiety (53%) and depression (47%).

Our findings are in line with recent national prevalence rates of mental distress in adolescents after the Beirut port explosion [23]. This study, conducted a few years post-blast and post-pandemic, illustrates the chronicity and lasting impact of accumulating adversities on adolescents. Although alarming, those findings might be expected in the backdrop of a prolonged collective challenges and traumas altering the quality of life, of the Lebanese people in general and on the most vulnerable youngsters in particular [50]. These crises are aggravated by the absence of governmental policies addressing mental health and the scarcity of school-based mental health interventions promoting wellbeing and interpersonal coping skills. As such, results should inform altogether clinicians, researchers, and policymakers to implement culturally sensitive strategies aiming at addressing wellbeing as well as managing anxious and depressive symptomatology among Lebanese adolescents to prevent potential worsening of mental and physical health.

Per se, the psychometric properties of the WHO-5 construct validity were examined using CFA, an approach systematically supported by validation researchers as an essential step in scale validation [47,51]. Our analyses showed the fit of a

one-factor model, replicating the single-factor structure of the original scale version [52]. Similarly, subsequently, psycho-metric studies across different countries [5] and different age groups including Arab adults and adolescents [16,18] have consistently supported the unidimensional structure. This could infer that the assessment of adolescents' wellbeing in Lebanon could be accounted for by a single underlying factor, comparably to age-matched participants from other countries. It mostly supports the applicability of the WHO-5 within the Lebanese setting as a unidimensional measure of wellbeing and its recommendation for use in Arab adolescents globally. Future studies could additionally investigate whether current findings within the general population could be replicated in clinical populations of adolescents, whereby a single latent trait would also help evaluate their wellbeing.

Moreover, the unidimensional factor structure of this Arabic version of the WHO-5 was identical for both male and female high schoolers at the three levels of invariance (Configural, Metric, and Scalar). We found no significant gender differences in scores, and as such our results are in support of indiscriminately using the WHO-5 across genders, aligning with findings from previous studies [9]. The WHO-5 items also appeared to be internally consistent, and quite similar to values obtained in a recent validation of the same Arabic version of WHO-5 in a sample of non-clinical Arab adults across six Arab countries including Lebanon [8]. One supplementary distinctive feature of this study was its evaluation of test-retest interclass correlation of wellbeing measures since more than half of the original sample of adolescents were retested at 3 months post-initial assessment. This test across time and across individuals although scarcely used in psychological measures allows to further refine the reliability of the scale by examining consistency and reproducibility [30]. Our results point to statistically adequate values and further evidence of the reliable psychometric properties of WHO-5 in Arab adolescents.

Lastly, the WHO-5 scores showed a strong and negative correlation with depression and anxiety scores. These findings also concur with prior research comparing the correlations between the WHO-5 with various measures of mental distress and related problems worldwide and more precisely in Middle Eastern and neighboring countries [8,53,54].

### Implications, limitations and recommendations for future research

Our results suggest that the Arabic version of the WHO-5 wellbeing index can be a valuable tool for researchers and clinicians aiming to assess wellbeing among Arab-speaking adolescents. This scale offers a quick, convenient, and age-appropriate means of assessing wellbeing, with the distinct advantage of being culturally sensitive and easy to understand. In addition, recent studies have highlighted that the WHO-5 could also serve as a proxy measure for clinical depression in the general population [55,56]. As such, psychologists, public health nurses, and educators could use it to assess wellbeing, and, by the same token, depressive symptoms among high schoolers, particularly those living in vulnerable environments.

This study supports the use of the WHO-5 Arabic version in young adolescents in a Middle Eastern country. Although psychometric properties indicate valid and reliable usage of the scale in a culturally sensitive manner, results should be generalized with care to other Arabic-speaking adolescents, especially since the sample included in the study was educated urban youth. Further research should as such target larger sample sizes and non-schooled youth. Additional limitation stems from relying solely on self-report measures that may introduce known biases such as social desirability, memory inaccuracies, and potential misinterpretations of items. Convergent validity was not assessed in this study in the absence of another scale to measure wellbeing in our survey. Nonetheless, the test-retest solid reliability, gender invariance, and religious as well as regional diversity representation within the targeted public schools in Lebanon could mitigate such biases. Future studies should further investigate the potential influence of other demographic factors on wellbeing, incorporate physiological markers, and potentially study clinical samples for a comprehensive understanding.

### Conclusion

Findings from this study have demonstrated that the psychometric properties of the Arabic WHO-5 wellbeing index in Lebanese adolescents are scientifically sound, supporting the usage of this easy-to-use, fast assessment tool in research.

This scale is highly efficient for assessing wellbeing in adolescents. At the individual level, it would allow an easy, reliable, and valid evaluation of wellbeing, a core construct with strong correlations to improved mental and physical health, increased resilience, better academic performance, and stronger social relationships. On the social versant, it would be relevant to track overall levels of wellbeing in larger samples to prevent and reduce healthcare costs, enhance productivity, and foster a more stable thriving society, especially in times of political instability and ongoing turmoil.

## Supporting information

**S1 Table. Validated Arabic Scales.** Items for WHO-5, PHQ-9 and GAD-7.
(DOCX)

**S2 Table. Validated Scales.** Items for WHO-5, PHQ-9 and GAD-7.
(DOCX)

## Acknowledgments

The authors wish to acknowledge all those who made the YEY project possible.

## Author contributions

**Conceptualization:** Myriam El Khoury Malhame, Maria-Jose Sanchez-Ruiz, Rita Doumit.

**Data curation:** Souheil Hallit, Maria-Jose Sanchez-Ruiz.

**Formal analysis:** Souheil Hallit.

**Funding acquisition:** Myriam El Khoury Malhame, Rita Doumit.

**Investigation:** Myriam El Khoury Malhame.

**Methodology:** Souheil Hallit, Maria-Jose Sanchez-Ruiz, Rita Doumit.

**Project administration:** Maria-Jose Sanchez-Ruiz.

**Software:** Souheil Hallit.

**Supervision:** Myriam El Khoury Malhame, Souheil Hallit.

**Visualization:** Rita Doumit.

**Writing – original draft:** Myriam El Khoury Malhame, Rita Doumit.

**Writing – review & editing:** Souheil Hallit, Maria-Jose Sanchez-Ruiz.

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
