## [Decision Letter · Decision Letter 0]

PONE-D-24-60467Psychometric validation of an Arabic version of the WHO-5 wellbeing scale among Lebanese AdolescentsPLOS ONE

Dear Dr. El Khoury Malhame,

Thank you for submitting your manuscript to PLOS ONE. After careful consideration, we feel that it has merit but does not fully meet PLOS ONE’s publication criteria as it currently stands. Therefore, we invite you to submit a revised version of the manuscript that addresses the points raised during the review process.

We look forward to receiving your revised manuscript.

Kind regards,

Janna Metzler

Academic Editor

PLOS ONE

Journal Requirements:

“This research was funded by Grand Challenges Canada – Global Mental Health – NIHS; R-GMH-POC-2107-47498.”

4. Please note that funding information should not appear in the Acknowledgments section or other areas of your manuscript. We will only publish funding information present in the Funding Statement section of the online submission form. Please remove any funding-related text from the manuscript. 

5. We note that you have indicated that there are restrictions to data sharing for this study. For studies involving human research participant data or other sensitive data, we encourage authors to share de-identified or anonymized data. However, when data cannot be publicly shared for ethical reasons, we allow authors to make their data sets available upon request. For information on unacceptable data access restrictions, please see http://journals.plos.org/plosone/s/data-availability#loc-unacceptable-data-access-restrictions. 

6. For studies involving third-party data, we encourage authors to share any data specific to their analyses that they can legally distribute. PLOS recognizes, however, that authors may be using third-party data they do not have the rights to share. When third-party data cannot be publicly shared, authors must provide all information necessary for interested researchers to apply to gain access to the data. (https://journals.plos.org/plosone/s/data-availability#loc-acceptable-data-access-restrictions) 

7. Please amend your authorship list in your manuscript file to include author “Sleiman El Hajj”. 

8. Your ethics statement should only appear in the Methods section of your manuscript. If your ethics statement is written in any section besides the Methods, please delete it from any other section. 

Reviewers' comments:

Reviewer's Responses to Questions

**Comments to the Author**

1. Is the manuscript technically sound, and do the data support the conclusions?

Reviewer #1: Partly

Reviewer #2: Yes

Reviewer #3: Yes

2. Has the statistical analysis been performed appropriately and rigorously? 

Reviewer #1: No

Reviewer #2: Yes

Reviewer #3: No

3. Have the authors made all data underlying the findings in their manuscript fully available?

Reviewer #1: No

Reviewer #2: Yes

Reviewer #3: No

4. Is the manuscript presented in an intelligible fashion and written in standard English?

Reviewer #1: Yes

Reviewer #2: Yes

Reviewer #3: Yes

5. Review Comments to the Author

Reviewer #1: Thank you for the opportunity to review this manuscript, which investigates the psychometric properties (i.e., factor structure, internal consistency, test-retest reliability, and convergent validity) of an Arabic version of the WHO-5 Well-Being Index among a sample of Lebanese adolescents.

I have three major concerns:

1. The study's contribution is somewhat unclear.

2. Previous psychometric findings of the WHO-5 in adolescent samples are not sufficiently discussed.

3. Some of the analyses could be expanded with more in-depth investigations.

Please see my detailed comments below.

Title: The scale is typically referred to as the "WHO-5 Well-Being Index" rather than "WHO-5 wellbeing scale". I recommend adjusting the title (and throughout the manuscript) accordingly to ensure accessibility for interested readers.

Abstract: In the abstract, write out "WHO-5 Well-Being Index" in full when first mentioned, followed by the acronym in parentheses (i.e., WHO-5).

Introduction

There is insufficient discussion of previous research on the psychometric properties of the WHO-5. Please provide more information on studies that employed adolescent samples. See for instance, Adjorlolo & Anum (2021), Allgaier et al. (2012), Blom et al. (2012), Cosma et al. (2022), and Sischka et al. (2025). Discussing these findings will help contextualize your study.

Methods

Closely adhere to survey reporting guidelines (e.g., Turk et al., 2018, supplemental material S2). The following methodological details should be specified:

- Data Collection Method: Was the survey conducted online or via postal mail?

- Survey Accessibility: If online, was it open to everyone or restricted (e.g., password-protected)?

- Recruitment Process: How was the survey advertised?

- Incentives: Were any offered?

- Question Requirements: If online, did the survey include mandatory questions (e.g., Sischka et al., 2022)?

- Item Order: If online, were the items randomized or presented in a fixed order?

How was the sample size determined? Please provide details regarding your sample size considerations (see Giner-Sorolla et al., 2024).

You wrote: "Students were asked about their age and gender, then filled out the validated, publicly available Arabic version of the standardized scales for well-being (WHO-5), depression (PHQ-9), and anxiety (GAD-7), in that order." If this version of the WHO-5 has already been validated (which aspects?), please provide references supporting this claim. Additionally, clarify what the contribution of the current study is.

You wrote “Some participants were tested twice for the same scales […]”. How were these participants selected? Was the second assessment available to all participants, or did only some opt to complete it?

In the Analytical Strategy section, you state: "There were no missing responses in the dataset." However, in the Participants section, you wrote: "The sample consisted of 701 high school students […]. Among those participants, 681 students had complete data and were included in the final sample […]." This suggests that missing data were handled through listwise deletion. If this is the case, please state it explicitly. As missing data can be effectively handled using full information maximum likelihood (FIML) in AMOS – which is widely regarded as a best-practice method (e.g., Newman, 2014) – I strongly recommend employing this approach instead.

The English WHO-5 response category reads as "At no time," but you wrote "Not at all." Please confirm whether this is an intended modification or a translation inconsistency.

Please include the Arabic translation of the WHO-5 (i.e., items, response categories, and instructions) in the manuscript.

Besides gender invariance, consider testing for age-group invariance as well.

Please specify how the latent variable was scaled in the CFA model (e.g., Little et al., 2006).

Add an explanation of the ICC in the Analytical Strategy section. Clearly specify which ICC coefficient was used, as different types exist with distinct assumptions and purposes (e.g., Aldridge et al., 2017).

Results

Consider providing local model fit information in addition to global fit statistics (e.g., Kline, 2024). This would also allow for comparison with previous WHO-5 psychometric studies, many of which have reported high correlated residuals for items 1 and 2.

Would it be possible to calculate effect size measures for measurement invariance (e.g., Nye et al., 2019; Gun et al., 2020)?

Please calculate item-level test-retest reliability coefficients (e.g., DeSimone, 2015) to provide a more nuanced view of item stability over time.

Discussion

The Results section states McDonald’s Omega = .83, but the Discussion section states McDonald’s Omega = .93. Please verify and correct this inconsistency.

Minor points

- You wrote “Pearson test was used to correlate the WHO-5 scores with the other scales […]”. Please revise this to: "Pearson correlations were calculated to examine the association between the WHO-5 scores and the other scales."

- Please change the heading “Interclass Correlation” to “Intraclass Correlation”.

This study addresses an important topic, and the manuscript is generally well-structured. However, clarifying the study's contribution, ensuring consistency in reporting, and expanding key analyses (e.g., local model fit, test-retest at the item level, measurement invariance effect sizes) would significantly strengthen the paper. Additionally, adherence to survey reporting guidelines is crucial for transparency and reproducibility.

I appreciate the authors' efforts and look forward to seeing the revised version.

References

Adjorlolo, S., & Anum, A. (2021). Positive and negative psychosis risk symptoms among adolescents in Ghana. International Journal of Adolescence and Youth, 26(1), 307–320. https://doi.org/10.1080/02673843.2021.1933110

Aldridge, V. K., Dovey, T. M., & Wade, A. (2017). Assessing test-retest reliability of psychological measures. European Psychologist, 22(4), 207-218. https://doi.org/10.1027/1016-9040/a000298

Allgaier, A.-K., Pietsch, K., Frühe, B., Prast, E., Sigl-Glöckner, J., & Schulte-Körne, G. (2012). Depression in pediatric care: Is the WHO-Five Well-Being Index a valid screening instrument for children and adolescents? General Hospital Psychiatry, 34(3), 234–241. https://doi.org/10.1016/j.genhosppsych.2012.01.007

Blom, E. H., Bech, P., Högberg, G., Larsson, J. O., & Serlachius, E. (2012). Screening for depressed mood in an adolescent psychiatric context by brief self-assessment scales - testing psychometric validity of WHO–5 and BDI-6 indices by latent trait analyses. Health and Quality of Life Outcomes, 10(1), 149. https://doi.org/10.1186/1477-7525-10-149

Cosma, A., Költö, A., Chzhen, Y., Kleszczewska, D., Kalman, M., & Martin, G. (2022). Measurement invariance of the WHO–5 Well-Being Index: Evidence from 15 European countries. International Journal of Environmental Research and Public Health, 19(16), 9798. https://doi.org/10.3390/ijerph19169798

DeSimone, J. A. (2015). New techniques for evaluating temporal consistency. Organizational Research Methods, 18(1), 133-152. https://doi.org/10.1177/1094428114553061

Giner-Sorolla, R., Montoya, A. K., Reifman, A., Carpenter, T., Lewis Jr, N. A., Aberson, C. L., ... & Soderberg, C. (2024). Power to detect what? Considerations for planning and evaluating sample size. Personality and Social Psychology Review, 28(3), 276-301.

Gunn, H. J., Grimm, K. J., & Edwards, M. C. (2020). Evaluation of six effect size measures of measurement non-invariance for continuous outcomes. Structural Equation Modeling, 27(4), 503-514. https://doi.org/10.1080/10705511.2019.1689507

Kline, R. B. (2024). How to evaluate local fit (residuals) in large structural equation models. International Journal of Psychology, 59(6), 1293-1306. https://doi.org/10.1002/ijop.13252

Little, T. D., Slegers, D. W., & Card, N. A. (2006). A non-arbitrary method of identifying and scaling latent variables in SEM and MACS models. Structural Equation Modeling, 13(1), 59-72. https://doi.org/10.1207/s15328007sem1301_3

Newman, D. A. (2014). Missing data: Five practical guidelines. Organizational Research Methods, 17(4), 372-411. https://doi.org/10.1177/1094428114548590

Nye, C. D., Bradburn, J., Olenick, J., Bialko, C., & Drasgow, F. (2019). How big are my effects? Examining the magnitude of effect sizes in studies of measurement equivalence. Organizational Research Methods, 22(3), 678-709.

Sischka, P. E., Martin, G., Residori, C., Hammami, N., Page, N., Schnohr, C., & Cosma, A. (2025). Cross-National Validation of the WHO-5 Well-Being Index Within Adolescent Populations: Findings From 43 Countries. Assessment. Advance online publication. https://doi.org/10.1177/10731911241309452

Sischka, P. E., Décieux, J. P., Mergener, A., Neufang, K. M., & Schmidt, A. F. (2022). The impact of forced answering and reactance on answering behavior in online surveys. Social Science Computer Review, 40(2), 405-425. https://doi.org/10.1177/0894439320907067

Turk, T., Elhady, M. T., Rashed, S., Abdelkhalek, M., Nasef, S. A., Khallaf, A. M., ... & Huy, N. T. (2018). Quality of reporting web-based and non-web-based survey studies: What authors, reviewers and consumers should consider. PloS one, 13(6), e0194239. https://doi.org/10.1371/journal.pone.0194239

Reviewer #2: Thanks for sending the paper. I see that this validations study is more focus on statistical basis (CFA). However, there are other forms of validity and reliability. Please see the paper.

Arafat, S. Y., Chowdhury, H. R., Qusar, M. M. A. S., & Hafez, M. A. (2016). Cross cultural adaptation and psychometric validation of research instruments: a methodological review. Journal of Behavioral Health, 5(3), 129-136.

Would you please check It??

Reviewer #3: I strongly suggest that the authors considers the addition of a descriptive table to show the sample profile, using variables of individual characteristics, such as gender, age, sexual orientation and grade level.

Regarding the analytic strategy, I strongly recommend the use of R or JASP to run the analysis. Furthermore, considering that the WHO-5 is a 6 point likert scale, I suggest to evaluate the multivariate normality assumption of SEM with Mardia’s test, and in case of violations in multivariate normality, I suggest to use the Weighted Least Squares Mean and Variance adjusted estimator (WLSMV), due to the ordinal nature of the items, instead of non-parametric bootstraps.

In the “Further analyses” topic, I suggest to use the Shapiro-Wilks or Kolmogorov-Smirnov test to assess the assumption of normality distribution, in order to use the t-test and the Pearson’s correlation index.

It seems that the authors know the relevance of assumptions of the statistical analysis, but the statistical procedure lacks on the correction of the violations of these assumptions.

The “Participants” topic is duplicated (it appears on methods and results), and it seems that this section in the “results” is referring to the test-retest reliability results. It’s strongly recommended to use the Student’s t test, Generalized Linear Models (GLM) or Mann-Whitney tests to compare groups, in order to assess whether the results are different between measurement times. The test-retest reliability can be tested using linear correlation between the results (pre and post), with strong correlations indicating good results.

In “confirmatory factor analysis of the WHO-5 scale”, the article mentions the composite reliability using the alpha and omega letters. It seems to be a typing error, these are not composite reliability indexes, but internal consistency indexes, and composite reliability is another one, calculated using factor loadings, it needs to be corrected.

Regarding the “gender invariance”, it’s not clear why to compare groups using t test by gender, since the factorial invariance is measured using CFA models, and not the general score of the instrument (table 2 is the correct for evaluating factorial invariance).

I suggest the authors to avoid very small topics, as interclass correlation and concurrent validity.

In the discussion, the authors mentions the “divergent validity”, but it's not clear how it was measured

6. PLOS authors have the option to publish the peer review history of their article (what does this mean? ). If published, this will include your full peer review and any attached files.

**Do you want your identity to be public for this peer review?** For information about this choice, including consent withdrawal, please see our Privacy Policy .

Reviewer #1: No

Reviewer #2: No

Reviewer #3: No

---

## [Author Response · Author response to Decision Letter 1]

1 May 2025

Dearest Editorial team

On behalf of my colleagues, we are grateful for the scientific rigorous comments provided to help uplift our manuscript and meet journal criteria

We are hereby providing a point-by-point reply to editor and reviewers in bold, and have highlighted in yellow tracked changes in the manuscript for ease of tracking

We look forward for the overall evaluation and are happy to be given the opportunity to give visibility to our work from the underrepresented part of the world.

Editor:

Done

Done

“This research was funded by Grand Challenges Canada – Global Mental Health – NIHS; R-GMH-POC-2107-47498.”

Done

4. Please note that funding information should not appear in the Acknowledgments section or other areas of your manuscript. We will only publish funding information present in the Funding Statement section of the online submission form. Please remove any funding-related text from the manuscript.

Noted. This has been processed accordingly.

5. We note that you have indicated that there are restrictions to data sharing for this study. For studies involving human research participant data or other sensitive data, we encourage authors to share de-identified or anonymized data. However, when data cannot be publicly shared for ethical reasons, we allow authors to make their data sets available upon request. For information on unacceptable data access restrictions, please see http://journals.plos.org/plosone/s/data-availability#loc-unacceptable-data-access-restrictions.

a) If there are ethical or legal restrictions on sharing a de-identified data set, please explain them in detail (e.g., data contain potentially identifying or sensitive patient information, data are owned by a third-party organization, etc.) and who has imposed them (e.g., a Research Ethics Committee or Institutional Review Board, etc.). Please also provide contact information for a data access committee, ethics committee, or other institutional body to which data requests may be sent

The authors do not have the right to share any data information since the data is owned by a third-party organization, the Lebanese American University. Data can be shared upon a reasonable request from Ms. Perla Zgheib (perla.z.zgheib@gmail.com), Lecturer, Department of Social and Education Sciences, School of Arts and Sciences, Lebanese American University, Beirut, Lebanon

Human Ethics approval and Consent to participate declaration. In accordance with the

Declaration of Helsinki, all participants and their parents approved and signed the

informed consent to participate. Adolescent participants and their parents signed a

consent form. The document was sent home to their parents to sign it while

adolescents signed their consent form in class.

The consent was given by the Institutional Review Board (LAU IRB; reference number:

LAU.SAS.MJ1.28/Oct/2021).

6. For studies involving third-party data, we encourage authors to share any data specific to their analyses that they can legally distribute. PLOS recognizes, however, that authors may be using third-party data they do not have the rights to share. When third-party data cannot be publicly shared, authors must provide all information necessary for interested researchers to apply to gain access to the data. (https://journals.plos.org/plosone/s/data-availability#loc-acceptable-data-access-restrictions)

7. Please amend your authorship list in your manuscript file to include author “Sleiman El Hajj”.

We apologize for the mishap; Dr Sleiman El Hajj is not an author on the manuscript.

8. Your ethics statement should only appear in the Methods section of your manuscript. If your ethics statement is written in any section besides the Methods, please delete it from any other section.

Done.

DONE. Thank you for the meticulous followup

Reviewer #1:

Thank you for the opportunity to review this manuscript, which investigates the psychometric properties (i.e., factor structure, internal consistency, test-retest reliability, and convergent validity) of an Arabic version of the WHO-5 Well-Being Index among Lebanese adolescents.

I have three major concerns:

1. The study's contribution is somewhat unclear.

The contribution of the study is to validate the use of a widely documented WHO-5 in an adolescent Arab speaking population. It is stated in the introduction “However, despite its widespread use, there is a surprising gap in validation studies within the Arab region, which is known to host one of the most youthful age structures globally”.

2. Previous psychometric findings of the WHO-5 in adolescent samples are not sufficiently discussed.

Noted. As per the suggestion, after listing some of the countries with validates WHO-5 in adolescents, we have discussed the psychometric findings “The scale has systematically shown a one factor structure similarly to the original version in adults with adequate convergent and divergent validity.”

3. Some of the analyses could be expanded with more in-depth investigations.

We changed the analyses and expanded the interpretation of the results. Please let us know if there is something else missing. Thank you.

Please see my detailed comments below.

Title: The scale is typically referred to as the "WHO-5 Well-Being Index" rather than "WHO-5 wellbeing scale". I recommend adjusting the title (and throughout the manuscript) accordingly to ensure accessibility for interested readers.

Thank you for the highlight, the naming has been amended in the title and the manuscript.

Abstract: In the abstract, write out "WHO-5 Well-Being Index" in full when first mentioned, followed by the acronym in parentheses (i.e., WHO-5).

Done.

Introduction

There is insufficient discussion of previous research on the psychometric properties of the WHO-5. Please provide more information on studies that employed adolescent samples. See for instance, Adjorlolo & Anum (2021), Allgaier et al. (2012), Blom et al. (2012), Cosma et al. (2022), and Sischka et al. (2025). Discussing these findings will help contextualize your study.

Thank you for the references. Some of these studies were added to better contextualize the current findings.

Methods

Closely adhere to survey reporting guidelines (e.g., Turk et al., 2018, supplemental material S2). The following methodological details should be specified:

- Data Collection Method: Was the survey conducted online or via postal mail?

The survey was conducted in person, in class for school students. This has been added accordingly.

- Survey Accessibility: If online, was it open to everyone or restricted (e.g., password-protected)?

- Question Requirements: If online, did the survey include mandatory questions (e.g., Sischka et al., 2022)?

- Item Order: If online, were the items randomized or presented in a fixed order?

To answer all three comments, we have clearly re-stated that the surveys were collected in class in person (and not online) and that the item order was fixed.

- Recruitment Process: How was the survey advertised?

The schools were randomly assigned by the Ministry of Education and the students within the appropriate age range were subsequently approached. This has been clarified in the manuscript

- Incentives: Were any offered?

No. No incentives were offered.

- How was the sample size determined? Please provide details regarding your sample size considerations (see Giner-Sorolla et al., 2024).

We have already included this information in the analytic strategy paragraph:

“We aimed to enroll a minimum of 100 adolescents following the recommendations of Mundfrom et al. of 3 to 20 times the number of the scale’s variables”

We have now included it in a separate paragraph for better clarity.

Giner-Sorolla et al. emphasize the importance of aligning sample sizes with the effect sizes researchers aim to detect. For CFA, this approach involves conducting a tailored power analysis that accounts for the number of factors and anticipated factor loadings. Given the brevity of the WHO-5 scale (five items loading on one factor), a sample size of 100 participants if often deemed sufficient to have enough statistical power (which is the same number we have already mentioned). We added the new reference in our paper and updated the text as follows:

We aimed to enroll a minimum of 100 adolescents following the recommendations of Mundfrom et al. of 3 to 20 times the number of the scale’s variables (30) and those of Giner-Sorolla et al (31).

- You wrote: "Students were asked about their age and gender, then filled out the validated, publicly available Arabic version of the standardized scales for well-being (WHO-5), depression (PHQ-9), and anxiety (GAD-7), in that order." If this version of the WHO-5 has already been validated (which aspects?), please provide references supporting this claim. Additionally, clarify what the contribution of the current study is.

The WHO-5 scale has already been validated in Arabic but in adult populations. The reference has already been used and was again provided and highlighted in the detailing of the WHO-5 scale.

- You wrote “Some participants were tested twice for the same scales […]”. How were these participants selected? Was the second assessment available to all participants, or did only some opt to complete it?

Only some opted to complete it, without any specific way to select those participants.

- In the Analytical Strategy section, you state: "There were no missing responses in the dataset." However, in the Participants section, you wrote: "The sample consisted of 701 high school students […]. Among those participants, 681 students had complete data and were included in the final sample […]." This suggests that missing data were handled through listwise deletion. If this is the case, please state it explicitly. As missing data can be effectively handled using full information maximum likelihood (FIML) in AMOS – which is widely regarded as a best-practice method (e.g., Newman, 2014) – I strongly recommend employing this approach instead.

We removed the sentence “There were no missing responses in the dataset”.

We added the results as follows:

Seven hundred adolescents participated in this study. The mean and standard deviation of the scores were as follows: WHO-5 wellbeing index (pre: 41.75 ± 22.61, post: 39.68 ± 21.24), PHQ-9 depression (11.10 ± 6.76), and GAD-7 anxiety (9.62 ± 5.50). The description of the categorization of participants according to their PHQ-9 and GAD-7 scores is summarized in “S1 Table”.

Table 1. Sociodemographic information and categorization of participants according to their depression and anxiety scores (N = 700)

Age (years) 16.27 ± 1.11

Gender

Males 295 (42.1%)

Females 378 (54.0%)

Depression severity

No depression (PHQ-9 scores 0-4) 135 (19.3%)

Mild (PHQ-9 scores 5-9) 193 (27.6%)

Moderate (PHQ-9 scores 10-14) 146 (20.9%)

Moderately severe (PHQ-9 scores 15-19) 130 (18.6%)

Severe (PHQ-9 scores 20-27) 95 (13.6%)

Anxiety severity

No anxiety (GAD-7 scores 0-4) 147 (21.0%)

Mild (GAD-7 scores 5-9) 226 (32.3%)

Moderate (GAD-7 scores 10-14) 174 (24.9%)

Severe (GAD-7 scores ≥15) 153 (21.9%)

*Numbers might not round to the total N because of missing values.

We analyzed the database using the software R now.

- The English WHO-5 response category reads as "At no time," but you wrote "Not at all." Please confirm whether this is an intended modification or a translation inconsistency.

This is an inattention error, the scale was used with the “At no time” category indeed.

- Please include the Arabic translation of the WHO-5 (i.e., items, response categories, and instructions) in the manuscript.

This has been included in the supplement material S4 alongside with all available Arabic scales

The English version is included in S5

- Besides gender invariance, consider testing for age-group invariance as well.

We analyzed age invariance as requested by the reviewer. Results are added in table 2.

- Please specify how the latent variable was scaled in the CFA model (e.g., Little et al., 2006).

We added the following sentence to the analytic strategy paragraph:

“The latent variable was scaled in the CFA model by fixing the loading of the first item to 1.”

- Add an explanation of the ICC in the Analytical Strategy section. Clearly specify which ICC coefficient was used, as different types exist with distinct assumptions and purposes (e.g., Aldridge et al., 2017).

We added the following lines to the analytic strategy paragraph:

In order to assess the test-retest reliability of the WHO-5 scale, we calculated the intraclass correlation coefficient (ICC) using the reliability analysis option in SPSS v.27 to ensure the scale’s stability over time. We used the two-way mixed-effect model ICC. The ICC values vary between 0 and 1, with values >0.75 considered excellent.

Reference: Aldridge, V. K., Dovey, T. M., & Wade, A. (2017). Assessing test-retest reliability of psychological measures: Persistent methodological problems. European Psychologist, 22(4), 207–218. https://doi.org/10.1027/1016-9040/a000298

Results

- Consider providing local model fit information in addition to global fit statistics (e.g., Kline, 2024). This would also allow for comparison with previous WHO-5 psychometric studies, many of which have reported high correlated residuals for items 1 and 2.

We added the following to the results section:

To evaluate local fit, we examined the modification indices and residuals; the correlation between items 1 and 2 was borderline (= 10.05); therefore, we did not add a correlation between those two items.

- Would it be possible to calculate effect size measures for measurement invariance (e.g., Nye et al., 2019; Gun et al., 2020)?

We added the value of the ef

---

## [Editor Report · Decision Letter 1]

PONE-D-24-60467R1Psychometric validation of an Arabic version of the WHO-5 wellbeing index among Lebanese AdolescentsPLOS ONE

Dear Dr. El Khoury Malhame,

Thank you for submitting your manuscript to PLOS ONE. After careful consideration, we feel that it has merit but does not fully meet PLOS ONE’s publication criteria as it currently stands. Therefore, we invite you to submit a revised version of the manuscript that addresses the points raised during the review process.

We look forward to receiving your revised manuscript.

Kind regards,

Janna Metzler

Academic Editor

PLOS ONE

**Additional Editor Comments:**

Thank you for your revision. I've read through your response to reviewers and supporting documents. To better facilitate the review process, it would be helpful to add page and line numbers for the revision and directly cite the page/line numbers in your response. This will allow our reviewers to go directly and confirm the information stated in the response. At a minimum, the revision should include a tracked changes version indicating text changes (rather than highlighted in yellow in the submitted documents) would be preferable prior to proceeding with the review process.

---

## [Author Response · Author response to Decision Letter 2]

10 May 2025

Dearest Editorial team

On behalf of my colleagues, we are grateful for the scientific rigorous comments provided to help uplift our manuscript and meet journal criteria

We are hereby providing a point-by-point reply to editor and reviewers in bold and have highlighted in yellow tracked changes in the manuscript for ease of tracking.

As per the editorial request, we have additionally added line numbering in the manuscript and in the responses to further smoothen the process.

We look forward for the overall evaluation and are happy to be given the opportunity to give visibility to our work from the underrepresented part of the world.

Editor:

Done

Done

“This research was funded by Grand Challenges Canada – Global Mental Health – NIHS; R-GMH-POC-2107-47498.”

Done on Page 1 lines 37-39

4. Please note that funding information should not appear in the Acknowledgments section or other areas of your manuscript. We will only publish funding information present in the Funding Statement section of the online submission form. Please remove any funding-related text from the manuscript.

Noted. This has been processed accordingly.

5. We note that you have indicated that there are restrictions to data sharing for this study. For studies involving human research participant data or other sensitive data, we encourage authors to share de-identified or anonymized data. However, when data cannot be publicly shared for ethical reasons, we allow authors to make their data sets available upon request. For information on unacceptable data access restrictions, please see http://journals.plos.org/plosone/s/data-availability#loc-unacceptable-data-access-restrictions.

a) If there are ethical or legal restrictions on sharing a de-identified data set, please explain them in detail (e.g., data contain potentially identifying or sensitive patient information, data are owned by a third-party organization, etc.) and who has imposed them (e.g., a Research Ethics Committee or Institutional Review Board, etc.). Please also provide contact information for a data access committee, ethics committee, or other institutional body to which data requests may be sent

The authors do not have the right to share any data information since the data is owned by a third-party organization, the Lebanese American University. Data can be shared upon a reasonable request from Ms. Perla Zgheib (perla.z.zgheib@gmail.com), Lecturer, Department of Social and Education Sciences, School of Arts and Sciences, Lebanese American University, Beirut, Lebanon

Human Ethics approval and Consent to participate declaration. In accordance with the

Declaration of Helsinki, all participants and their parents approved and signed the

informed consent to participate. Adolescent participants and their parents signed a

consent form. The document was sent home to their parents to sign it while

adolescents signed their consent form in class.

The consent was given by the Institutional Review Board (LAU IRB; reference number:

LAU.SAS.MJ1.28/Oct/2021).

6. For studies involving third-party data, we encourage authors to share any data specific to their analyses that they can legally distribute. PLOS recognizes, however, that authors may be using third-party data they do not have the rights to share. When third-party data cannot be publicly shared, authors must provide all information necessary for interested researchers to apply to gain access to the data. (https://journals.plos.org/plosone/s/data-availability#loc-acceptable-data-access-restrictions)

7. Please amend your authorship list in your manuscript file to include author “Sleiman El Hajj”.

We apologize for the mishap; Dr Sleiman El Hajj is not an author on the manuscript. This has been amended and all authors have sent their approval regarding the change of authorship by email reply to the editorial office.

8. Your ethics statement should only appear in the Methods section of your manuscript. If your ethics statement is written in any section besides the Methods, please delete it from any other section.

Done.

DONE. Thank you for the meticulous follow-up.

Reviewer #1:

Thank you for the opportunity to review this manuscript, which investigates the psychometric properties (i.e., factor structure, internal consistency, test-retest reliability, and convergent validity) of an Arabic version of the WHO-5 Well-Being Index among Lebanese adolescents.

I have three major concerns:

1. The study's contribution is somewhat unclear.

The contribution of the study is to validate the use of a widely documented WHO-5 in an adolescent Arab speaking population since none of the age group nor the cultural validity were previously assessed. It is rephrased more clearly in the introduction (P.4, L9-11) “However, despite its widespread use, there is a surprising gap in validation studies within the Arab region, which is known to host one of the most youthful age structures globally”.

2. Previous psychometric findings of the WHO-5 in adolescent samples are not sufficiently discussed.

Noted. As per the suggestion, after listing some of the countries with validates WHO-5 in adolescents, we have discussed the psychometric findings (P.3, L3 and P.4 L1-2) “In those single country studies, the WHO-5 wellbeing index has systematically shown a one factor structure similarly to the original version in adults with adequate convergent and divergent validity.”

3. Some of the analyses could be expanded with more in-depth investigations.

We changed as per the recommendation:

- the analyses (P.7-P.8 L3)

- and expanded the interpretation of the results (P9, L1-7, P.9 L12-14, P.10, L1-7 and L13-15).

Please let us know if there is something else missing. Thank you.

Please see my detailed comments below.

Title: The scale is typically referred to as the "WHO-5 Well-Being Index" rather than "WHO-5 wellbeing scale". I recommend adjusting the title (and throughout the manuscript) accordingly to ensure accessibility for interested readers.

Thank you for the highlight, the naming has been amended in the title and the manuscript P.1, L1.

Abstract: In the abstract, write out "WHO-5 Well-Being Index" in full when first mentioned, followed by the acronym in parentheses (i.e., WHO-5).

Done on page 2, line 8.

Introduction

There is insufficient discussion of previous research on the psychometric properties of the WHO-5. Please provide more information on studies that employed adolescent samples. See for instance, Adjorlolo & Anum (2021), Allgaier et al. (2012), Blom et al. (2012), Cosma et al. (2022), and Sischka et al. (2025). Discussing these findings will help contextualize your study.

Thank you for the references. Some of these studies were added to better contextualize the current findings in the introduction/background (P.4, L1-9) and in the discussion (P.11, L26 and P.12, L18-19).

Methods

Closely adhere to survey reporting guidelines (e.g., Turk et al., 2018, supplemental material S2). The following methodological details should be specified:

- Data Collection Method: Was the survey conducted online or via postal mail?

The survey was conducted in person, in class for school students. This has been added accordingly (P.5, L22).

- Survey Accessibility: If online, was it open to everyone or restricted (e.g., password-protected)?

- Question Requirements: If online, did the survey include mandatory questions (e.g., Sischka et al., 2022)?

- Item Order: If online, were the items randomized or presented in a fixed order?

To answer all three comments, we have clearly re-stated that the surveys were collected in class in person (and not online) and that the item order was fixed (P.5, L22-25).

- Recruitment Process: How was the survey advertised?

The schools were randomly assigned by the Ministry of Education and the students within the appropriate age range were subsequently approached. This has been clarified in the manuscript (P. 5, L16-18).

- Incentives: Were any offered?

No. No incentives were offered.

- How was the sample size determined? Please provide details regarding your sample size considerations (see Giner-Sorolla et al., 2024).

This information was previously defined in the analytic strategy paragraph:

“We aimed to enroll a minimum of 100 adolescents following the recommendations of Mundfrom et al. of 3 to 20 times the number of the scale’s variables”

We have now included it in a separate paragraph on “Minimal sample size” for better clarity (P.5, L27-P.6, L3).

Giner-Sorolla et al. emphasize the importance of aligning sample sizes with the effect sizes researchers aim to detect. For CFA, this approach involves conducting a tailored power analysis that accounts for the number of factors and anticipated factor loadings. Given the brevity of the WHO-5 scale (five items loading on one factor), a sample size of 100 participants if often deemed sufficient to have enough statistical power (which is the same number we have already mentioned).

As such, we added the new reference in our paper and updated the text as follows:

We aimed to enroll a minimum of 100 adolescents following the recommendations of Mundfrom et al. of 3 to 20 times the number of the scale’s variables (30) and those of Giner-Sorolla et al (31).

- You wrote: "Students were asked about their age and gender, then filled out the validated, publicly available Arabic version of the standardized scales for well-being (WHO-5), depression (PHQ-9), and anxiety (GAD-7), in that order." If this version of the WHO-5 has already been validated (which aspects?), please provide references supporting this claim. Additionally, clarify what the contribution of the current study is.

The WHO-5 scale has already been validated in Arabic but in adult populations. The reference has already been used and was again provided and highlighted in the detailing of the WHO-5 scale (P.4, L9-12).

- You wrote “Some participants were tested twice for the same scales […]”. How were these participants selected? Was the second assessment available to all participants, or did only some opt to complete it?

Only some opted to complete it, without any specific way to select those participants. This was clarified (P.5, L19-20).

- In the Analytical Strategy section, you state: "There were no missing responses in the dataset." However, in the Participants section, you wrote: "The sample consisted of 701 high school students […]. Among those participants, 681 students had complete data and were included in the final sample […]." This suggests that missing data were handled through listwise deletion. If this is the case, please state it explicitly. As missing data can be effectively handled using full information maximum likelihood (FIML) in AMOS – which is widely regarded as a best-practice method (e.g., Newman, 2014) – I strongly recommend employing this approach instead.

We removed the sentence “There were no missing responses in the dataset” and instead of the initial deletion we used the recommended approach. The database was analyzed using the R software (P.7, L1-4).

We revisited the analyses accordingly, with slight changes as per the above. The results participants section now reads as follows (P.8, L6-10) and the below table was amended accordingly:

Seven hundred adolescents participated in this study. The mean and standard deviation of the scores were as follows: WHO-5 wellbeing index (pre: 41.75 ± 22.61, post: 39.68 ± 21.24), PHQ-9 depression (11.10 ± 6.76), and GAD-7 anxiety (9.62 ± 5.50). The description of the categorization of participants according to their PHQ-9 and GAD-7 scores is summarized in “S1 Table”.

Table 1. Sociodemographic information and categorization of participants according to their depression and anxiety scores (N = 700)

Age (years) 16.27 ± 1.11

Gender

Males 295 (42.1%)

Females 378 (54.0%)

Depression severity

No depression (PHQ-9 scores 0-4) 135 (19.3%)

Mild (PHQ-9 scores 5-9) 193 (27.6%)

Moderate (PHQ-9 scores 10-14) 146 (20.9%)

Moderately severe (PHQ-9 scores 15-19) 130 (18.6%)

Severe (PHQ-9 scores 20-27) 95 (13.6%)

Anxiety severity

No anxiety (GAD-7 scores 0-4) 147 (21.0%)

Mild (GAD-7 scores 5-9) 226 (32.3%)

Moderate (GAD-7 scores 10-14) 174 (24.9%)

Severe (GAD-7 scores ≥15) 153 (21.9%)

*Numbers might not round to the total N because of missing values.

- The English WHO-5 response category reads as "At no time," but you wrote "Not at all." Please confirm whether this is an intended modification or a translation inconsistency.

This is a typo error; the scale was used with the “At no time” category indeed. It was amended in the manuscript (P.6, L-9).

- Please include the Arabic translation of the WHO-5 (i.e., items, response categories, and instructions) in the manuscript.

This has been included in the supplement material S4 alongside with all available Arabic scales. The English version is included in S5.

- Besides gender invariance, consider testing for age-group invariance as well.

We analyzed age invariance as requested by the reviewer (P.7, L18-19).

Results were added in Table 2 (P.9, L16).

- Please specify how the latent variable was scaled in the CFA model (e.g., Little et al., 2006).

We added the following sentence to the analytic strategy paragraph (P.7, L10):

“The latent variable was scaled in the CFA model by fixing the loading of the first item to 1.”

- Add an explanation of the ICC in the Analytical Strategy section. Clearly specify which ICC coefficient was used, as different types exist with distinct assumptions and purposes (e.g., Aldridge et al., 2017).

We added the following lines to the analytic strategy paragraph (P.7, L27-29 till P.8, L1-3):

“In order to assess the test-retest reliability of the WHO-5 scale, we cal

---

## [Decision Letter · Decision Letter 2]

PONE-D-24-60467R2Psychometric validation of an Arabic version of the WHO-5 wellbeing index among Lebanese AdolescentsPLOS ONE

Dear Dr. El Khoury Malhame,

Thank you for submitting your manuscript to PLOS ONE. After careful consideration, we feel that it has merit but does not fully meet PLOS ONE’s publication criteria as it currently stands. Therefore, we invite you to submit a revised version of the manuscript that addresses the points raised during the review process.

The manuscript has been improved through changes made on the most recent revision. However, there are still methodological considerations (see Reviewer 1 comments) that require clarification prior to acceptance.Prior to acceptance, the manuscript should include the following changes: 1. The estimation methods must be clearly specified.2. Explicitly state the ICC model used and whether your aim was to assess consistency or absolute agreement.3. Clarify whether the internal consistency of each instrument cited in the manuscript were calculated using the current sample or drawn from previous studies.

We look forward to receiving your revised manuscript.

Kind regards,

Janna Metzler

Academic Editor

PLOS ONE

Reviewers' comments:

Reviewer's Responses to Questions

**Comments to the Author**

1. If the authors have adequately addressed your comments raised in a previous round of review and you feel that this manuscript is now acceptable for publication, you may indicate that here to bypass the “Comments to the Author” section, enter your conflict of interest statement in the “Confidential to Editor” section, and submit your "Accept" recommendation.

Reviewer #1: (No Response)

Reviewer #2: All comments have been addressed

Reviewer #3: All comments have been addressed

2. Is the manuscript technically sound, and do the data support the conclusions?

Reviewer #1: Partly

Reviewer #2: Yes

Reviewer #3: Yes

3. Has the statistical analysis been performed appropriately and rigorously? 

Reviewer #1: Yes

Reviewer #2: Yes

Reviewer #3: Yes

4. Have the authors made all data underlying the findings in their manuscript fully available?

Reviewer #1: No

Reviewer #2: Yes

Reviewer #3: No

5. Is the manuscript presented in an intelligible fashion and written in standard English?

Reviewer #1: No

Reviewer #2: Yes

Reviewer #3: Yes

6. Review Comments to the Author

Reviewer #1: Thank you for the opportunity to re-review this manuscript. Some aspects have been improved, but there are still several areas that require further consideration.

In the Methods section, three different CFA estimation methods are mentioned (i.e., WLSMV, maximum likelihood, robust maximum likelihood; p. 7, lines 5, 6, and 14). Please clarify which estimator was actually used and revise the description accordingly to accurately reflect your analytic procedure. Misreporting estimation methods can cause confusion and undermine the reproducibility of your results.

Regarding effect sizes, I was referring to measurement invariance effect sizes that indicate the magnitude of non-invariance (e.g., Gunn et al., 2020; Nye et al., 2019), not effect sizes for mean differences. That said, this was only a suggestion.

The phrase "two-way mixed-effect model ICC" is not specific enough to determine exactly which ICC type was used. SPSS provides several ICC variants under this model, depending on whether one is assessing single vs. average measures, and consistency vs. absolute agreement (see also Shrout & Fleiss, 1979). To enhance transparency and allow proper interpretation, please explicitly state the ICC model used (e.g., ICC(3,1)) and whether your aim was to assess consistency or absolute agreement. This level of detail is important for evaluating the reliability of the WHO-5 across time points.

Please also clarify whether the internal consistency estimates for the GAD-7 and PHQ-9 reported in the Measures section were calculated using the current sample, or if they were taken from previous studies. If the latter is the case, please additionally report the internal consistency values based on your current sample.

The manuscript should be thoroughly proofread, as some sentences remain awkwardly phrased and/or vague/ambiguous. A few examples:

- “However, some missing data was found in the other variables (age, gender, etc.) and were not replaced.” Please rephrase into something like “Some missing data were present in demographic variables such as age and gender. These missing values were not imputed and were handled using listwise deletion in the relevant analyses.”

- “Normality was not verified since the p value of the Shapiro-Wilk test turned out significant (p < 0.001).” Please rephrase into something like “The distribution of the WHO-5 scores was not normal, as indicated by a significant result from the Shapiro-Wilk test (p < .001).”

References

Aldridge, V. K., Dovey, T. M., & Wade, A. (2017). Assessing test-retest reliability of psychological measures: Persistent methodological problems. European Psychologist, 22(4), 207–218. https://doi.org/10.1027/1016-9040/a000298

Gunn, H. J., Grimm, K. J., & Edwards, M. C. (2020). Evaluation of six effect size measures of measurement non-invariance for continuous outcomes. Structural Equation Modeling, 27(4), 503-514. https://doi.org/10.1080/10705511.2019.1689507

Nye, C. D., Bradburn, J., Olenick, J., Bialko, C., & Drasgow, F. (2019). How big are my effects? Examining the magnitude of effect sizes in studies of measurement equivalence. Organizational Research Methods, 22(3), 678-709. https://doi.org/10.1177/1094428118761122

Shrout, P. E., & Fleiss, J. L. (1979). Intraclass correlations: Uses in assessing rater reliability. Psychological Bulletin, 86(2), 420–428. https://doi.org/10.1037/0033-2909.86.2.420

Reviewer #2: Psychometric validation of an Arabic version of the WHO-5 wellbeing index among Lebanese Adolescents" (PONE-D-24-60467R2)

Thanks

Reviewer #3: (No Response)

7. PLOS authors have the option to publish the peer review history of their article (what does this mean? ). If published, this will include your full peer review and any attached files.

**Do you want your identity to be public for this peer review?** For information about this choice, including consent withdrawal, please see our Privacy Policy .

Reviewer #1: No

Reviewer #2: No

Reviewer #3: **Yes: ** Julio Cezar Albuquerque da Costa

---

## [Author Response · Author response to Decision Letter 3]

6 Jun 2025

Dearest Editor,

Thank you and the team for the rigorous followup

We are hereby providing our comments and point-by-point review in bold below. Changes are highlighted in the manuscript to facilitate tracking

We do hope the uplifted version suits the publication criteria of PlosOne.

Very much looking forward to the publication

Myriam EL khour-Malhame

Dear Dr. El Khoury Malhame,

Thank you for submitting your manuscript to PLOS ONE. After careful consideration, we feel that it has merit but does not fully meet PLOS ONE’s publication criteria as it currently stands. Therefore, we invite you to submit a revised version of the manuscript that addresses the points raised during the review process.

The manuscript has been improved through changes made on the most recent revision. However, there are still methodological considerations (see Reviewer 1 comments) that require clarification prior to acceptance.

Prior to acceptance, the manuscript should include the following changes:

1. The estimation methods must be clearly specified.

Parameter estimates were obtained using the robust maximum likelihood method.

2. Explicitly state the ICC model used and whether your aim was to assess consistency or absolute agreement.

We used the ICC(3,1), i.e., two-way mixed-effect model ICC, consistency type and single measures.

This was clarified in the statistical analysis paragraph.

3. Clarify whether the internal consistency of each instrument cited in the manuscript were calculated using the current sample or drawn from previous studies.

We look forward to receiving your revised manuscript.

Kind regards,

Janna Metzler

Academic Editor

PLOS ONE

Reviewer #1:

Thank you for the opportunity to re-review this manuscript. Some aspects have been improved, but there are still several areas that require further consideration.

- In the Methods section, three different CFA estimation methods are mentioned (i.e., WLSMV, maximum likelihood, robust maximum likelihood; p. 7, lines 5, 6, and 14). Please clarify which estimator was actually used and revise the description accordingly to accurately reflect your analytic procedure. Misreporting estimation methods can cause confusion and undermine the reproducibility of your results.

We apologize for the confusion. We kept this sentence:

Parameter estimates were obtained using the robust maximum likelihood method.

- Regarding effect sizes, I was referring to measurement invariance effect sizes that indicate the magnitude of non-invariance (e.g., Gunn et al., 2020; Nye et al., 2019), not effect sizes for mean differences. That said, this was only a suggestion.

We apologize for the misunderstanding. We added the results as advised according to the references suggested. Thank you.

- The phrase "two-way mixed-effect model ICC" is not specific enough to determine exactly which ICC type was used. SPSS provides several ICC variants under this model, depending on whether one is assessing single vs. average measures, and consistency vs. absolute agreement (see also Shrout & Fleiss, 1979). To enhance transparency and allow proper interpretation, please explicitly state the ICC model used (e.g., ICC(3,1)) and whether your aim was to assess consistency or absolute agreement. This level of detail is important for evaluating the reliability of the WHO-5 across time points.

SPSS provides several models: two-way mixed, two-way random, and one-way random.

For the type, it can be either consistency or absolute agreement.

We used the ICC(3,1), i.e., two-way mixed-effect model ICC, consistency type and single measures.

This was clarified in the statistical analysis paragraph.

- Please also clarify whether the internal consistency estimates for the GAD-7 and PHQ-9 reported in the Measures section were calculated using the current sample, or if they were taken from previous studies. If the latter is the case, please additionally report the internal consistency values based on your current sample.

We added the Cronbach’s alpha values in this study.

The manuscript should be thoroughly proofread, as some sentences remain awkwardly phrased and/or vague/ambiguous. A few examples:

- “However, some missing data was found in the other variables (age, gender, etc.) and were not replaced.” Please rephrase into something like “Some missing data were present in demographic variables such as age and gender. These missing values were not imputed and were handled using listwise deletion in the relevant analyses.”

Sentence rephrased as suggested by the reviewer.

- “Normality was not verified since the p value of the Shapiro-Wilk test turned out significant (p < 0.001).” Please rephrase into something like “The distribution of the WHO-5 scores was not normal, as indicated by a significant result from the Shapiro-Wilk test (p < .001).”

Sentence rephrased as suggested by the reviewer.

References

Aldridge, V. K., Dovey, T. M., & Wade, A. (2017). Assessing test-retest reliability of psychological measures: Persistent methodological problems. European Psychologist, 22(4), 207–218. https://doi.org/10.1027/1016-9040/a000298

Gunn, H. J., Grimm, K. J., & Edwards, M. C. (2020). Evaluation of six effect size measures of measurement non-invariance for continuous outcomes. Structural Equation Modeling, 27(4), 503-514. https://doi.org/10.1080/10705511.2019.1689507

Nye, C. D., Bradburn, J., Olenick, J., Bialko, C., & Drasgow, F. (2019). How big are my effects? Examining the magnitude of effect sizes in studies of measurement equivalence. Organizational Research Methods, 22(3), 678-709. https://doi.org/10.1177/1094428118761122

Shrout, P. E., & Fleiss, J. L. (1979). Intraclass correlations: Uses in assessing rater reliability. Psychological Bulletin, 86(2), 420–428. https://doi.org/10.1037/0033-2909.86.2.420

Thank you for sending the references. We have incorporated them in our paper.

We hope the revised version is up to your expectations. Thank you again for your time and efforts.

---

## [Decision Letter · Decision Letter 3]

Psychometric validation of an Arabic version of the WHO-5 wellbeing index among Lebanese Adolescents

PONE-D-24-60467R3

Dear Dr. El Khoury Malhame,

We’re pleased to inform you that your manuscript has been judged scientifically suitable for publication and will be formally accepted for publication once it meets all outstanding technical requirements.

Kind regards,

Janna Metzler

Academic Editor

PLOS ONE

Additional Editor Comments (optional): The following comments were provided by reviewer #1. Should you wish to edit the text further, do let me know. 

Reviewers' comments:

Reviewer's Responses to Questions

**Comments to the Author**

1. If the authors have adequately addressed your comments raised in a previous round of review and you feel that this manuscript is now acceptable for publication, you may indicate that here to bypass the “Comments to the Author” section, enter your conflict of interest statement in the “Confidential to Editor” section, and submit your "Accept" recommendation.

Reviewer #1: All comments have been addressed

2. Is the manuscript technically sound, and do the data support the conclusions?

Reviewer #1: Yes

3. Has the statistical analysis been performed appropriately and rigorously? 

Reviewer #1: Yes

4. Have the authors made all data underlying the findings in their manuscript fully available?

Reviewer #1: No

5. Is the manuscript presented in an intelligible fashion and written in standard English?

Reviewer #1: Yes

6. Review Comments to the Author

Reviewer #1: Thank you for the opportunity to re-review this manuscript. All remaining substantive comments have been adequately addressed, and I appreciate the authors’ thoughtful revisions. However, the manuscript would still benefit from a careful language review. Several sentences remain awkwardly phrased or imprecise, which affects readability. To improve clarity and polish, I strongly recommend a thorough language revision—ideally by a native English speaker or a professional academic proofreader. If that is not feasible, you may carefully use a language model such as ChatGPT (https://chatgpt.com/, see e.g., https://scitechedit.com/chatgpt-and-academic-writing/ point 4 and 5) to help revise the text. Please note that such tools should be used critically and cautiously—always reviewing the output to ensure it accurately reflects your intended meaning and maintains scientific precision. If you choose to use ChatGPT, I recommend revising the manuscript section by section using a prompt like the following: "Please help me revise the following section of an academic manuscript for grammar, clarity, and style. It should remain formal and scientifically accurate. Avoid changing the meaning, but feel free to rephrase awkward or unclear sentences. Here is the text: [Insert section here]."

As a brief note on style: I often provide numerous references in my reviews. These are intended to guide and inform the authors and should not necessarily be cited in the manuscript unless I explicitly recommend doing so.

7. PLOS authors have the option to publish the peer review history of their article (what does this mean? ). If published, this will include your full peer review and any attached files.

**Do you want your identity to be public for this peer review?** For information about this choice, including consent withdrawal, please see our Privacy Policy .

Reviewer #1: **Yes: ** Philipp E. Sischka

---

## [Editor Report · Acceptance letter]

PONE-D-24-60467R3

PLOS ONE

Dear Dr. El Khoury Malhame,

I'm pleased to inform you that your manuscript has been deemed suitable for publication in PLOS ONE. Congratulations! Your manuscript is now being handed over to our production team.

Kind regards,

on behalf of

Dr Janna Metzler

Academic Editor

PLOS ONE